# Peer-Delivered Hepatitis C Testing and Health Screening Provided in a Community Pharmacy Setting: Proof of Concept

**DOI:** 10.3390/pharmacy13060154

**Published:** 2025-10-27

**Authors:** Neha Sankla, Ray Cottington, Chris Cowie, Paul Huggett, Leila Reid, Stuart Smith, Sorcha Daly, Danny Morris, James Spear, Amanda Marsden, David Richards, Rachel Halford, Scott Walter, Jenny Scott

**Affiliations:** 1Population Health Sciences, Bristol Medical School, University of Bristol, Bristol BS8 2PS, UK; 2Hepatitis C Trust, 72 Weston Street, London SE1 3HQ, UK; 3ODN Leicester & Northampton Hepatitis C Network, University Hospitals Leicester NHS Trust, Leicester LE1 5WW, UK; 4Gastroenterology Department, Northampton General Hospital, Northampton NN1 5BD, UK; 5National Institute for Health and Care Research, Applied Research Collaboration West (NIHR ARC West), University Hospitals Bristol and Weston NHS Foundation Trust, Bristol BS1 2NT, UK; 6Health Economics and Health Policy, Bristol Medical School, University of Bristol, Bristol BS8 1NU, UK; 7Centre for Academic Primary Care, Bristol Medical School, University of Bristol, Bristol BS8 2PS, UK

**Keywords:** hepatitis C testing, community pharmacy, health screening, primary care

## Abstract

In order to reach and maintain hepatitis C virus (HCV) elimination goals, it is imperative to reach marginalized people who do not engage with traditional testing and treatment. Peer-led interventions are effective in engaging such individuals. Studies have demonstrated community pharmacy as a low-threshold setting for HCV testing, but pharmacy teams’ competing demands are a barrier to maximizing potential. This pilot project aimed to assess whether a pharmacy-based peer-led model of HCV testing was implementable, deliverable, able to engage marginalized people, and overcome pharmacy staff constraints. We implemented a peer-led HCV testing service in one community pharmacy in the Midlands, England, providing four focused phases of testing, totalling 198 h, over two years. In total, 591 tests for antibodies or RNA were undertaken, identifying 24 active infections. Subsequent phases retested 20%, 16%, and 11% of those tested in combined preceding phases. In response to feedback, phases 3 and 4 included health screening (blood pressure, cholesterol, diabetes, and nurse-performed Fibroscans^®^). We demonstrate engagement and the ability to identify and refer those with abnormal results to appropriate healthcare. This pilot shows that peer-led testing in the pharmacy setting can be implemented and warrants further scale up and evaluation.

## 1. Introduction

Globally, an estimated 50 million people have chronic hepatitis C virus (HCV) infection, with about 1 million new infections per year. In 2022, there were approximately 242,000 HCV-related deaths [1]. A global elimination target of an 80% reduction in new HCV infections has been set by the World Health Organization (WHO) for 2030, aiming to avert 7.1 million deaths [2]. Transmission primarily occurs through receipt of contaminated blood products and injection with contaminated equipment [1]. In the UK, considerable progress has been made to reduce HCV prevalence amongst people who inject drugs. In England, in 2023, the number of adults living with chronic HCV decreased by 56.7% from 2015 statistics [3]. This is due to very significant efforts made under a national HCV elimination programme, led by the National Health Service (NHS) England working with local NHS treatment teams, drug treatment services, lived experience organisations, prisons, the pharmaceutical industry, and harm reduction services. However, around 70% of people who inject drugs who are living with HCV are unaware of their infection [3]. To reach the WHO elimination goal, efforts need to be stepped up to reach these people who are both less likely to engage in testing and treatment and more likely to transmit HCV.

Community pharmacies in the UK have a long history of providing services for people who use drugs. Pharmacy needle and syringe provision (NSP) began in 1987 [4], and supervised consumption of opioid substitution therapies (OSTs) began in the 1990s [4,5]. These services are commissioned at the local level according to the need and willingness on the part of the pharmacy team to provide them. In England, community pharmacies follow a positive care law, and 99% of people living in high-deprivation areas are within a 20 min walk of a pharmacy [6]. Pharmacies tend to be open longer than other primary care or drug services. Their convenience and accessibility are important drivers of public choice to use them for healthcare services [7]. Pharmacies are therefore potentially well placed to offer HCV and other blood-borne virus testing.

HCV testing has been successfully demonstrated in community pharmacies internationally, in 24 studies, across 11 different countries, supporting both case detection and retention during treatment [8]. In 22 of these studies, testing was undertaken by pharmacy staff, 1 used in-reach nurses [9], and another used in-reach phlebotomists [10]. A national community pharmacy HCV testing pilot was launched in England in Sept 2020, during the COVID-19 vaccination programme. For a detailed description, see [11] (Figure 1). This pilot was open to any community pharmacy that met the requirements, which included having a consultation room. In practice, this is the majority of pharmacies. However, as the pilot targeted people who inject drugs not engaged with drug treatment services, in reality, only pharmacies that offer NSP (around 18% of pharmacies in England [12,13]) were likely to be able to identify and proactively offer the service to eligible people [14]. This pilot stopped on 31st March 2023. The data show that 1058 payment claims were made (1 claim = 1 test), across 17 Integrated Care Boards (local partnerships of health and social care organisations), largely concentrated in a small number of pharmacies [15]. These data also show relatively high claims in the first quarter of 2023 (349) compared to all four quarters of 2022 (672) and 2021 (21), suggesting that the service was developing. Staff pressures, including limited time during the COVID-19 vaccination programme, the narrow eligibility criteria, the process of antibody (as opposed to RNA to detect active infection) screening requiring onward referral, and the system of test kit ordering are thought to be barriers to this pilot and prevented the delivery of more widespread results [14]. Pharmacist time pressures, the expectation of a negative reaction from clients, and training needs are key barriers to public health service delivery more broadly [16]. A major barrier that people who use drugs experience or anticipate when accessing pharmacy services aimed at them is stigmatising attitudes amongst staff [17,18]. Ample research suggests that at least among some pharmacy staff, stigmatising attitudes are a legitimate concern (e.g., [19,20]).

Peer-led interventions have been shown to reduce health inequalities [21]. They have improved outcomes in mental healthcare [22]. Peer support has been shown to be particularly effective for marginalised people, including those with low literacy, supporting engagement and retention in healthcare [23]. There are no peer models in Hayes et al.’s 2023 systematic review of community pharmacy-based HCV testing [8]. The Hepatitis C Trust (HCT) is a peer-led UK charity; the majority of its staff and volunteers have lived experience relevant to hepatitis C. The HCT has developed a peer support model which increased the proportion of people initiating HCV treatment (RR 1.12 95%, 1.02–1.21) and completing treatment (OR 2.45 95%, 1.49–3.84) compared to non-peer support models [24]. The HCT model engages people in testing and offers advice on reducing risk, often through outreach. To leverage the accessibility of community pharmacies and address barriers like staff time constraints and stigma, the HCT adapted their peer-led model to the pharmacy setting. This model incorporates best practice from the I-COPTIC (Implementation of community pharmacy-based testing for hepatitis C) consensus statement on pharmacy HCV testing [14]. This includes offering point-of-care RNA testing to anyone with antibodies and widening access as outlined below. The long-term goal is to develop a replicable, scalable model enabling peer teams to work through appropriate pharmacies in a given locality, supporting testing and treatment among NSP and OST clients, effectively “micro-eliminating” HCV in each setting and ensuring this is sustained by returning every 6–12 months.

In this paper, we aimed to understand whether the HCT peer-led pharmacy model was implementable and deliverable as proof of concept. The HCT introduced it in one community pharmacy with existing NSP and OST services, aiming to effectively reach and engage people who use drugs in pharmacy-based HCV testing. As the service evolved, the provision of basic health screening alongside HCV testing was added in response to feedback, because this population tends not to engage in primary care services [25,26]. In this service evaluation, we report HCV testing and health screening engagement and the results from one pharmacy over four blocks of intervention. We also summarise service user feedback on improvements and pharmacy staff thoughts on enablers.

## 2. Materials and Methods

The peer-led model was implemented in one community pharmacy operating a busy NSP and OST dispensing service in one town in England, over four phases between September 2021 and September 2024. Phase 1 ran for five separate days (08:30 to 16:00), plus four evening shifts (16:00 to 20:00) between September 2021 and February 2022. The three subsequent phases each took place over five consecutive days, with Phase 2 in February 2023, Phase 3 in March 2024, also including four evening shifts, and Phase 4 in September 2024, including three evening shifts. The peer-led team consisted of HCT peer workers, HCT peer volunteers, and an NHS Trust community engagement lead. For Phases 2, 3, and 4, there was also an NHS Trust Viral Hepatitis Clinical Nurse Specialist. Onward referral pathways were agreed with the local hospital Hepatology Department. Participants were recruited opportunistically as they attended the pharmacy for NSP or OST services. Accompanying persons and other members of the public who enquired were also included in Phase 4, if after explaining HCV transmission, they perceived themselves at risk. Peers and pharmacy staff worked together to identify and proactively engage NSP and OST clients, and then show them to the private consultation room, or if occupied, the mobile clinic van parked outside. In both cases, peers explained the process, obtained informed consent, and performed the tests. The peers were trained and experienced in performing the tests and held NHS honorary contracts. A GBP 5 voucher was given to participants as contingency management to promote participation.

Consented individuals underwent an oral swab to test for markers of previous HCV infection (antibodies). Those who tested positive for HCV antibodies underwent ribonucleic acid (RNA) dry blood spot testing (DBST) to check current infection status. In the following cases, participants underwent DBST only: a previous HCT client known to have had HCV infection, including those known to have previously self-cleared infection; known to have had treatment and require testing for sustained virological response (SVR); and not known but self-declared previous infection. Those who showed active infection were referred for treatment and supported through this by the peer team. Appropriate harm reduction advice was given. An anonymous survey seeking feedback and views on service expansion was administered to willing participants during Phase 2, and as a result, peer-administered blood pressure (BP) measurements and liver Fibroscan^®^ screening by a nurse were added to Phase 3, and peer-administered cholesterol and diabetes screening was added to Phase 4. Those with a BP outside of the range 90/60 to 140/90 mmHg, total cholesterol above 5 mmol/L, and non-fasting blood sugar above 11 mmol/L were advised to see their GP. Those with a Fibroscan^®^ result < 7 kPa needed no action, results between 7 and 10 kPa were sent to the GP for review with advice on further testing, then subsequent referral to Hepatology if needed, and results > 10 kPa were referred directly to Hepatology as this indicates significant liver fibrosis/cirrhosis. Those with steatosis grade S1 or S2 (238–290 dB/m) were given lifestyle advice and the results sent to the GP; those with steatosis grade S3 (>290 dB/m) were given lifestyle advice, the results sent to the GP and the patient rebooked for nurse-led follow-up of repeat Fibroscan^®^ within 2 years. Those who were S0 (<238 dB/m) with any level of fibrosis were referred to Hepatology services by the nurse who conducted the scan. GPs played no other role in this pilot service. An overview of the pilot service is given in Figure 1:

**Figure 1 pharmacy-13-00154-f001:**
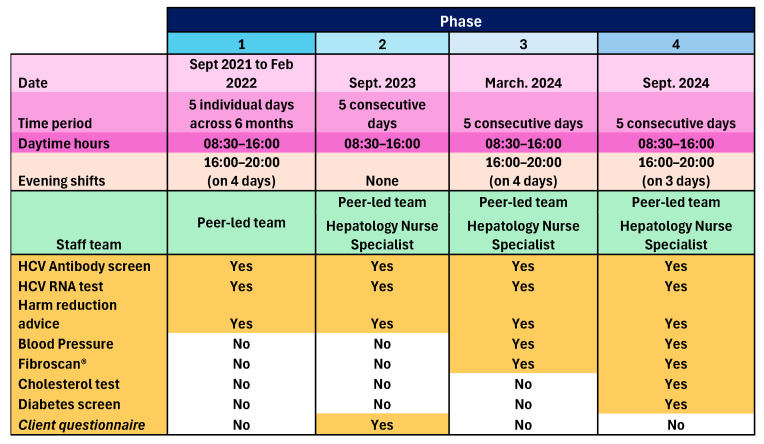
A summary of the four phases of the pilot service.

The pharmacy owner and duty pharmacist gave verbal feedback to JS on the factors they felt facilitated the success of the pilot.

Data were collated by the HCT peer team, overseen by RC and PH. Data analysis was undertaken by NS and JS with support from SW, LR, and RC. Before data receipt by analysts, participant names were removed from the spreadsheet and replaced with a code. Descriptive statistics were used to summarise participants, test uptake, test results, and subsequent actions using counts and proportions. Where the count was four or less, the exact number has been replaced with <5 in the reporting to protect anonymity. Feedback from the pharmacy owner and duty manager was documented in short note form, organised under categories, and used to inform our discussion of findings.

## 3. Results

### 3.1. HCV Testing

Across the four phases, 591 testing episodes were conducted over a total available service time of 198 h, identifying 24 active infections. In Phase 2, 33 people were retested who were initially tested in phase 1 (i.e., 20% of those tested in Phase 1 were retested in Phase 2). In Phase 3, 44 people were retested, having previously been tested in Phases 1 or 2 (16%), and in Phase 4, 47 people were retested, having been tested in Phases 1, 2, or 3 (11%).

In Phase 1, 12% of those who underwent DBST were found to have active infection. In Phases 2, 3, and 4, this figure was <5% each time. Active infections identified in Phase 4 were found in people who did not use the pharmacy for OST/NSP. HCT peers supported those who tested positive to engage in HCV treatment (see Table 1).

### 3.2. Feedback from Phase 2 Participants

Seventy-two people who took part in Phase 2 completed a feedback questionnaire. Responses were generally positive regarding the experience of the testing service, and their strength of preference for additional screening to be added alongside HCV testing was generally high, as shown in Table 2.

### 3.3. Health Screening

In response to the service evaluation feedback in Phase 2, peer-administered blood pressure checks and liver Fibroscans^®^ were included in Phase 3. A total of 68 (43%) people underwent a blood pressure check and 39 (24%) underwent a liver scan. In Phase 4, 51 (32%) people underwent a blood pressure check, while 65 (41%) underwent a liver scan. Phase 4 also included cholesterol testing, performed in 41 (26%), and blood glucose finger prick testing, performed in 27 (17%). Health screening is summarized in Table 3.

### 3.4. Pharmacist Feedback

Pharmacy owner and duty pharmacist views on what made the peer-led model successful from their perspective are summarized in Table 4.

## 4. Discussion

This evaluation of the pilot has demonstrated proof of concept, showing that it is possible to successfully implement the HCT model of peer-led testing in a pharmacy setting and deliver the service to individuals who use that pharmacy for substance use services (OST and NSP). Although not anticipated in advance, testing was also performed for some accompanying persons and other members of the public who expressed interest and identified themselves as at risk of HCV. This is an unforeseen advantage of pharmacy-based testing. The peer-led testing team operated over four phases in 2021, 2023, and 2024, totalling 20 testing days and 11 evenings (198 h). Phases 2, 3, and 4 were week-long blocks, while Phase 1 days were spread out. The numbers engaged in each phase were similar, except for Phase 2 which had lower recruitment. Winter time and the absence of evening sessions may have contributed, as well as the coinciding operation of another local initiative targeted at the same population, offering significantly greater incentives.

A total of 591 antibody and/or RNA tests were performed over the four phases, with a proportion in each subsequent phase(s) having been tested previously. The pharmacy context of regular, often daily client attendance lends itself to facilitating this repeat testing and could be particularly useful to check for reinfection after treatment. Across each phase, the proportion of those who had ever been infected declined. We note for Phase 4 that this figure is particularly low because antibody testing was targeted at those who had not previously tested positive. Those who had previously tested positive were given DBST only. The proportion with current active infection declined across the phases. The anonymised data did not link individual cases across the four phases so we are not able to report on longitudinal outcomes.

The HCT peer-led testing model involved between three and six peers at any one time, targeting as many people as possible, proactively and opportunistically when they attended the pharmacy for OST or NSP. Unlike pharmacists, peers were not constrained by other workflow demands such as dispensing, providing other services or responding to other customer or staff requests. Instead, peers could solely focus on engaging people in testing. Pharmacist-delivered models, even when supported by other pharmacy staff, depend on staff availability to offer and conduct the test at the time the person presents. This may require people being asked to wait. In a busy pharmacy environment, waiting is likely to impact testing completion rates. Waiting is an identified concern of people who use drugs when engaging in pharmacy services [27]. In contrast, the peer-led model is agile and can respond to people quickly. The use of the outreach mobile clinic van as a ‘spill over’ consultation room also prevented waiting. We also suggest that peer-led models are less likely to be hampered by feelings of anticipated stigma in the target group or enacted stigma from the provider, known barriers to pharmacy services for people who use drugs [17,28]. HCT peers also have extensive experience conducting outreach and engaging people who face high barriers to accessing healthcare; these skills strongly support the delivery of this intervention. Future research could confirm this.

It is important to emphasise that we are not arguing against pharmacist or pharmacy staff-led HCV testing. It is clear from a review by Hayes et al. that pharmacist-led testing and treatment can be successful and perform comparably to other community-based initiatives [8]. It is not possible to directly compare peer-led recruitment in our pilot with pharmacy staff recruitment in the studies within Hayes et al. [8], because some are multi-cite and report pooled data, and not all studies report the duration or number of hours of available testing. We can tentatively compare the Phase 1 results in our pilot with studies included in the sub-group meta-analysis by recruitment strategy reported by Hayes et al. [8], because we also used targeted recruitment towards people attending the pharmacy for OST and/or NSP. Our antibody positive rate was 44% and Hayes et al. [8] reported an average of 32.5%. Both can be considered very successful findings and evidence to support the use of targeted recruitment approaches within either peer-led or pharmacist-led pharmacy models of HCV testing. A limit in making this comparison is that some of the studies in Hayes et al.’s [8] meta-analysis targeted risk groups different to ours, including some targeted due to birth cohort and tattooing. We did not combine the results from our first three phases (which were all targeted recruitment) for comparison with Hayes et al. [8] because each subsequent phase included people who had been tested and treated in the pharmacy previously. The national community pharmacy HCV testing pilot in England (2020–2023) did not maximise opportunity for the reasons stated [11]. The present pilot study addressed these limits, by drawing on the I-COPTIC recommendations [14], having wider eligibility (people using the pharmacy for OST or NSP services), and offering the flexibility to include others who perceived themselves at risk. It also offered point-of-care RNA detection, with peer-supported onward referral into treatment. At present, community pharmacies in England cannot dispense HCV medications, but they are supplied by hospitals. The peers supported those with active infection to access hospital treatment, overcoming this potential hurdle. A modelling study has suggested that if testing and treatment efforts were stopped at the point of achieving elimination in England, up to 25% of people who inject drugs would be infected or reinfected with HCV within five years [29]. Therefore, ongoing efforts must be sustained to keep HCV at bay, and pharmacy has a valuable role to play in this [14]. We have shown that our peer-led testing model can enhance the scope of pharmacy-based testing by testing large numbers of people in relatively short, concentrated periods of time. By making a targeted effort to test as many people as possible who access the pharmacy for OST and NSP services, over repeated phases, the peer-led model shows the ability to both identify new infections and support local micro-elimination. The model has the potential to make a valuable contribution to sustaining micro-elimination, especially in areas of high deprivation, taking advantage of the close proximity of residents to community pharmacies [6].

The Phase 2 service evaluation feedback suggests that the service was well received, with 90% agreeing that the service was easy to access, private, and confidential. Most (93%) agreed that the test was well explained and they were given useful information on hepatitis C and/or harm reduction (86%). With regard to additional health checks and support, at least three-quarters of respondents to the Phase 2 survey were in favour of each of the suggested service expansions (Table 2). The limit of such tick box surveys is the lack of information on what is behind these views, and we acknowledge the possibility of responder bias due to the survey being given to them by a peer. The results tentatively suggest broad support for the peer-led model, but qualitative interviews to explore views and experiences in more depth would give richer data on how to develop the model for the future.

The feedback from the pharmacists highlights the importance of situating the model in pharmacies with a substantial existing target client group, enough space to comfortably operationalise, and the importance of positive pharmacy staff attitudes and a willingness to try new ways of working. Our team also endorse these views. These factors should be considered when identifying future pharmacies to base the peer-led service in.

The addition of health checks to the peer-led model was supported by 85% of people who answered the Phase 2 survey, with 80% supporting liver health checks. Between 17 and 41% of people who engaged in HCV testing participated in the health screening, demonstrating willingness and support. Our data provide evidence of engagement and show additional wider benefits to the peer-led model in the pharmacy setting. Prior engagement rates in preventative screening by our participants were not established. Engagement rates of people who use drugs in general are unreported, but are anticipated to be low. They face many barriers to accessing primary care, including homelessness, a lack of finance, and poor rapport with primary care staff [30]. They often avoid seeking treatment for health conditions until they become severe, due to fear of stigma and previous negative experiences with healthcare providers [30,31]. Enablers in primary care engagement include peer support models and having services co-located to other services that are salient, including drug treatment services [30]. One could reasonably suggest that this may also be true of co-location to OST dispensing and NSP services, such as community pharmacies, where clients often attend daily. Our findings suggest support for further work to establish whether a community pharmacy-based, peer-led model of health screening improves outcomes for people who use drugs. Such a model could expand, with appropriate training, staff skill-mix, and referral pathways, to address additional health concerns in this population, such as chronic obstructive pulmonary disease and ill mental health [30].

To the best of our knowledge, this is the first evidence to show that peer-led HCV testing can be successfully implemented in a community pharmacy setting. This model may be an effective way to overcome some of the barriers to community pharmacy testing already known, including pharmacy staff time and competing pressures [8,9] and client fear of stigma [17]. Attention to the I-COPTIC consensus on best practice [14] instead of the narrower NHS England pathway model [11] contributed to success. The availability of the mobile outreach van nearby was important, especially for pharmacies with limited available private consultation space.

One of the limits of the peer-led model we tested is that it makes no provision for those pharmacy users missed during the testing phases. A combination of peer-led intensive testing phases coupled with pharmacist-led testing available at other times may be a valuable approach in achieving and sustaining micro-elimination. This could be tested in future work. This pilot evaluation is limited by being conducted in one pharmacy only. Although we have demonstrated proof of concept, more work is needed to understand if the peer-led testing model would be successful in other pharmacies and whether it could be scaled up to support sustained micro-elimination and routine health screening. Such scale up should be further guided by I-COPTIC consensus statements [14]. Following up with future participants who take part in more than one phase would allow us to gain more understanding of the contribution of this model to sustaining elimination efforts. Testing the model in a range of pharmacy spatial configurations, with and without contingency management (e.g., monetary incentives), is needed. More needs to be done to understand how the service could be improved, its outcomes, and what leads to them. Qualitative insights from participants, peers, and pharmacists are needed. A realist evaluation methodology is advocated for studying further expansion of this model. In the future, research could compare the peer-led pharmacy model with other models of HCV testing and treatment in both pharmacy and other settings.

## 5. Conclusions

We have demonstrated the first peer-led model of hepatitis C testing with onward support into treatment and health screening in a community pharmacy setting. Future work is needed to demonstrate whether this model can be implemented at scale, and if so, future research is needed to compare the HCT peer-led model with other pharmacy and community-based models to quantify the contribution it can make to micro-elimination at the local level.

## Figures and Tables

**Table 1 pharmacy-13-00154-t001:** Characteristics of those tested and results from peer-led pharmacy-based Hepatitis C Trust testing programme.

Phase	1 September 2022	2 February 2023	3 March 2024	4 September 2024
**Total number of people who engaged in HCV testing** (antibodies +/or RNA screening)	169	102	160	160
**Gender**				
Male	129 (76%)	Not recorded	116 (73%)	Not recorded
Female	40 (24%)		44 (27%)	
**Age (years)**				
Less than 30	5 (3%)	12 (12%)	14 (9%)	12 (8%)
30–39	42 (25%)	28 (27%)	47 (29%)	38 (24%)
40–49	58 (35%)	37 (36%)	55 (34%)	66 (41%)
50 and over	18 (11%)	9 (9%)	44 (28%)	42 (26%)
Missing	46 (28%)	16 (17%)	0	<5
**HCV Risk Status**				
Current Injector	60 (35%)	26 (25%)	58 (36%)	31 (19%)
Past Injector	50 (29%)	28 (27%)	62 (39%)	52 (33%)
Never Injected	<5	39 (38%)	40 (25%)	77 (48%)
Missing/declined	60 (35%)	10 (9%)	0	0
**Pharmacy services currently accessible (can be both)**				
OST	144 (85%)	67 (65%)	114 (71%)	67 (42%)
NSP	43 (25%)	71 (68%)	62 (39%)	28 (18%)
**HCV screening results and outcomes**				
Tested positive for antibodies (ever infected) (as % of those tested)	73 (43%)	34 (33%)	30 (19%)	<5
RNA positive (active infection) (as % of those tested)	20 (12%)	<5	<5	<5
Referred to treatment within phase follow-up (lost to follow-up)	16 (4)	100%	100%	100%

Note: Counts less than 5 have been suppressed. Where necessary, counts have been rounded to prevent back calculation of suppressed counts.

**Table 2 pharmacy-13-00154-t002:** Phase 2 service evaluation self-administrated questionnaire results (*n* = 72).

Survey Questions	Agree *n* (%)	Unsure/No *n* (%)
1. The service easy to access.	65 (90%)	7 (10%)
2. The testing was comfortable, private, and confidential.	65 (90%)	7 (10%)
3. The staff explained the test to me well.	67 (93%)	5 (6%)
4. The staff gave me useful information about hepatitis C and/or harm reduction.	62 (86%)	10 (14%)
5. I would get tested here again.	67 (93%)	5 (6%)
6. I would recommend this service to others.	67 (93%)	5 (6%)
7. Would you like general health checks alongside your Hepatitis C testing?	61 (85%)	11 (15%)
8. Would you like liver health checks?	58 (80%)	14 (19%)
9. Would you like heart and body mass index checks?	59 (82%)	13 (18%)
10.Would you like vaccinations for Hepatitis B?	59 (82%)	13 (18%)
11. Would you like to be checked for other blood borne viruses (HIV/HBV)?	58 (81%)	14 (19%)
12. Would you like Naloxone provision as part of this service?	57 (79%)	15 (20%)
13. Would you like referral/help accessing a foodbank?	56 (78%)	16 (22%)
14. Would you like support or advice accessing housing?	56 (78%)	16 (22%)
15. Would you like support or advice accessing drug services?	55 (77%)	17 (23%)

**Table 3 pharmacy-13-00154-t003:** Health screening uptake and results (Phases 3 and 4 only).

Additional Health Checks	Phase 3 Number of People (% of Total Who Engaged in Phase)	Phase 4 Number of People (% of Total Who Engaged in Phase)
**Blood pressure (BP) measured by peers**	68 (43%)	51 (32%)
Referred to GP as BP outside 90/60 to 140/90 mmHg range	0	9 (6%)
**Liver fibrosis scan (Fibroscan^®^) by nurse**	39 (24%)	65 (41%)
Liver stiffness results range (kPa)	3.0 kPa–9.5 kPa	2.8 to 65.3 kPa
Referred to hepatology (see text)	<5	<5
Diagnosed with cirrhosis	0	<5
**Total cholesterol finger prick screening by peers**		41 (27%)
Referred to GP as total cholesterol above 5 mmol/L		8 (5%)
**Blood glucose finger prick testing by peers**		27 (17%)
Non-fasting blood glucose range (mmol/L)		* 3.4 to 9.9 mmol/L *
Referred to GP (above 11 mmol/L)		0

Note: counts less than 5 have been suppressed.

**Table 4 pharmacy-13-00154-t004:** Contextual factors that the pharmacy manager and duty pharmacist considered important facilitators in the success of the HCT pilot.

Pharmacy Factors	Staff Factors
Location in a deprived town centre area	Humility and a willingness to ‘go the extra mile’
Already high number of clients attending for OST and NSP	Forward thinking attitude, willing to try new ways of working
Internal space which enabled two consulting rooms to be made available for the HCT programme without interrupting other pharmacy activity	Non-judgmental attitude towards the client group
Parking directly outside pharmacy for the HCT mobile clinic van, which was used for part of the service (see Methods Section)	Past evidence of and confidence in creative service delivery (example given: COVID-19 vaccine outreach in religious spaces)

## Data Availability

The raw data supporting the conclusions of this article will be made available by the authors on request. (contact leila.reid@hepctrust.org.uk).

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
