# Peer review of "Peer-Delivered Hepatitis C Testing and Health Screening Provided in a Community Pharmacy Setting: Proof of Concept"

_pharmacy, 2025, doi:10.3390/pharmacy13060154_

Round 1

Reviewer 1 Report

Comments and Suggestions for Authors

Thank you kindly for your work on the peer-delivered hepatitis C testing manuscript.  I am truly impressed by your work and by your presentation of the data.  As a reviewer who frequently offers over 20 suggestions or comments, this manuscript is going to represent an outlier.  

Line  102 you use the acronym I-COPTIC and it has not been previously defined.  I realize that it is defined on line 381, but it should be defined here too.

Table 4, line 218 When reading the table I didn't understand that the parking for the van was important because the van was used for a part of the service.  It wasn't until I read the body of the text that I made the connection.  If this could be more clear, that would be good.

That's it, I only have 2 comments.

Well done

Author Response

Thank you for your kind comments, they are warmly appreciated in the harsh world of academia.

Comment: Line  102 you use the acronym I-COPTIC and it has not been previously defined.  I realize that it is defined on line 381, but it should be defined here too.

Response: Thank you for spotting this, we have added the definition in full (line 102-103). We have bracketed the definition as the statement is known as I-COPTIC.

Comment: Table 4, line 218 When reading the table I didn't understand that the parking for the van was important because the van was used for a part of the service.  It wasn't until I read the body of the text that I made the connection.  If this could be more clear, that would be good.

Response: Thank you for this comment, we have attempted to clarify by adding to the text in Table 4 (line 219). It now says 'Parking directly outside pharmacy for the HCT mobile clinic van, which was used for part of the service (see Methods section)'.

Reviewer 2 Report

Comments and Suggestions for Authors

Well documented proof of concept of a peer-delivered hepatitis C testing in a community pharmacy setting. I only have a few minor remarks:

I suggest that the abstract mentions the region where the experiment was held : for instance adding “… in one community pharmacy in England… ” in line 25. 

For the non native English speaker that I am, using the term “marginalized people” feels pejorative. If this is the case (the authors are better placed than me), it could be replace by “people who inject drugs” for instance.

Line 69 : reference is made to a figure 1, but I can’t fnd this in the manuscript

Line 138 : who received the voucher : the peers or the participants? Did it play any role in engagement?

Line 154-155 : referral to “their GP”. Does this mean the tested people have a GP? How were the GP's involved in recruting and testing?

Author Response

Comment: I suggest that the abstract mentions the region where the experiment was held : for instance adding “… in one community pharmacy in England… ” in line 25. 

Response: Thank you. We have added 'in the Midlands, England' to line 26.

Comment: For the non native English speaker that I am, using the term “marginalized people” feels pejorative. If this is the case (the authors are better placed than me), it could be replace by “people who inject drugs” for instance.

Response: Thank you for the comment. This term is commonly used in the UK to describe people marginalized from mainstream healthcare because it is not designed to meet their needs. It is not seen as pejorative, so we have chosen to keep it. As our participants were broader than injectors, I would not described them as such.

Comment: Line 69 : reference is made to a figure 1, but I can’t find this in the manuscript.

Response: This refers to figure 1 in the citation {11]. Apologies for any confusion, we have moved the square bracket around the citation to make it clear figure 1 is within citation number 11.

Comment: Line 138 : who received the voucher : the peers or the participants? Did it play any role in engagement?

Response: The voucher was given to participants. We have added this to line 138. It was intended to promote engagement, as stated (line 139).

Comment: Line 154-155 : referral to “their GP”. Does this mean the tested people have a GP? How were the GP's involved in recruiting and testing?

Response: Yes almost everyone in the UK has a GP, as registration with a GP is required to access other community services, including drug and alcohol services. However, this does not mean people access GP services. GPs were not involved in this work in any way, except they were notified of any liver results of concern as described (lines 156-160). This is standard practice across healthcare providers and GPs would be expected to try to engage the patient. We have amended line 154 to be clear participants were advised to see their GP about other results that were outside of normal range. We have added to line 163: GPs played no other role in this pilot service.

Reviewer 3 Report

Comments and Suggestions for Authors

The study presents an interesting concept and innovative approach; however, the manuscript would benefit from improved clarity and organization.

  1. Consider providing a visual diagram for the “four focused phases of testing,” illustrating the timeline, key activities, and objectives of each phase.
  2. The description of the pharmacy-based peer-led model should clarify potential differences if implemented outside a pharmacy setting, including any impact on accessibility, credibility, and resource availability.
  3. The discussion should more thoroughly compare peer-led and pharmacist-led models, highlighting their respective advantages, limitations, and potential for hybrid approaches.
  4. Including qualitative insights from peers and pharmacists could strengthen the discussion, particularly regarding feasibility, acceptability, and challenges in practice.
  5. Consider addressing implementation challenges and reliability, as well as the rationale and potential limitations of conducting the intervention in only a single community pharmacy.

Author Response

Comment: Consider providing a visual diagram for the “four focused phases of testing,” illustrating the timeline, key activities, and objectives of each phase.

Response: Thank you for this suggestion. They timeline and key activities are described in lines 124 - 130. The objectives are described in 111-119. As each phase evolved based on the previous, we feel it would be difficult to conceptualise in a diagram that adds any more clarity beyond what is already described.

Comment: The description of the pharmacy-based peer-led model should clarify potential differences if implemented outside a pharmacy setting, including any impact on accessibility, credibility, and resource availability.

Response: Thank you for this comment. There are many descriptions in the literature of peer-led HCV testing in other settings. However, it is not our intention to implement the peer-led model we describe elsewhere, it was developed specifically to fit with the pharmacy setting. We do not believe such a comparison fits with the aims of our pilot.

Comment: The discussion should more thoroughly compare peer-led and pharmacist-led models, highlighting their respective advantages, limitations, and potential for hybrid approaches.

Response: Thank you for this suggestion. With respect, our work is only a pilot and we feel to have adequate data to make comparisons between peer-led and pharmacist-led HCV testing, a full scale study would be needed. As ours is the first pharmacy based work to use peer-led testing, no such data exists. In line 349-351 we believe we address the need for this in future research where we say: 'In the future, research could compare the peer-led pharmacy model with other models of HCV testing and treatment in both pharmacy and other settings'.

Comment: Including qualitative insights from peers and pharmacists could strengthen the discussion, particularly regarding feasibility, acceptability, and challenges in practice.

Response: Thank you for this suggestion, we have added to line 347-348: . 'Qualitative insights from participants, peers and pharmacists are needed'.

Comment: Consider addressing implementation challenges and reliability, as well as the rationale and potential limitations of conducting the intervention in only a single community pharmacy.

Response: Thank you for this comment. We believe this is already addressed in the Discussion see lines 338 to 341, which say: 'This pilot evaluation is limited by being conducted in one pharmacy only. Although we have demonstrated proof of concept, more work is needed to understand if the peer-led testing model would be successful in other pharmacies and whether it could be scaled up to support sustained micro-elimination and routine health screening'. 

Round 2

Reviewer 1 Report

Comments and Suggestions for Authors

Thank you kindly for allowing me to review the updated version of your manuscript.  I admire the work you have done and are doing for patients who may have Hepatitis C.  

Your edits have made an improvement to your manuscript and I am satisfied with this presentation. 

Author Response

Thank you again for your review.

Reviewer 2 Report

Comments and Suggestions for Authors

Thank you for adressing the few questions I had

Author Response

Thank you again for your review.

Reviewer 3 Report

Comments and Suggestions for Authors

Most of the comments need to be addressed further.

Author Response

Thank you again for your review. We have made further revisions to the manuscript to address your concerns regarding the unresolved comments:

Comment 1: Consider providing a visual diagram for the “four focused phases of testing,” illustrating the timeline, key activities, and objectives of each phase.

Authors response: Figure 1 now added.

Comment 2: The description of the pharmacy-based peer-led model should clarify potential differences if implemented outside a pharmacy setting, including any impact on accessibility, credibility, and resource availability.

The model is specific to community pharmacy and has not been developed for use elsewhere and there is no intention to use elsewhere. The Hepatitis C Trust do have a model they use in other settings but we do not think a comparison is appropriate as the models differ and the goal of this paper is to highlight the potential of the model implemented in a pharmacy setting. If the reviewer means something different to our understanding, please do clarify.

Comment 3: The discussion should more thoroughly compare peer-led and pharmacist-led models, highlighting their respective advantages, limitations, and potential for hybrid approaches.

Please see the additional text which is blue track changes. The original manuscript covers the potential for hybrid approaches already.

Round 3

Reviewer 3 Report

Comments and Suggestions for Authors

I would like to thank the authors for the revisions made to the manuscript. The changes effectively address the concerns I raised in my previous review, and I appreciate the effort the authors have put into improving the clarity and depth of the content. The revised manuscript is now much stronger, and I am satisfied with the adjustments.
Therefore, I am pleased to recommend the manuscript for publication.